# FPGA Implementation of a Higher SFDR Upper DDFS Based on Non-Uniform Piecewise Linear Approximation

**Xuan Liao [1], Longlong Zhang [1], Xiang Hu [1]**  **, Yuanxi Peng [1] and Tong Zhou [2,\*]**

1. State Key Laboratory of High-Performance Computing, College of Computer Science and Technology, National University of Defense Technology, Changsha 410073, China; liaoxuan21@nudt.edu.cn
2. College of Advanced Interdisciplinary Studies, National University of Defense Technology, Changsha 410073, China
* Correspondence: zhoutong09@nudt.edu.cn

**Abstract:** We propose a direct digital frequency synthesizer (DDFS) by using an error-controlled piecewise linear (PWL) approximation method. For a given function and a preset max absolute error (MAE), this method iterates continuously from right to left within the input interval, dividing the entire interval into multiple segments. Within each segment, the least squares method is used to approximate the objective function, ensuring that each segment meets the error requirements. Based on this method, We first implemented a set of DDFS under different MAE to study the relationship between SFDR and MAE, and then evaluated its hardware overhead. In order to increase the frequency of the output signal, we implement a multi-core DDFS using time interleaving scheme. The experimental results show that our DDFS has significant advantages in SFDR, using fewer hardware resources to achieve high SFDR. Specifically, the SFDR of proposed DDFS can reach 114 dB using 399 LUTs, 66 flip flops and 3 DSPs. More importantly, we demonstrate through experiments that proposed DDFS breaks the SFDR theoretical upper bound of DDFS based on piecewise linear approximation methods.

**Keywords:** direct digital frequency synthesis;non-uniform piecewise linear approximation; spurious free dynamic range

## 1. Introduction

DDFS technology can generate the ideal sinusoid signal. Because of its advantages of fast frequency switching, high frequency resolution and low phase noise, it is widely used in modern communication and radar systems. Tierney et al. [1] first proposed DDFS architecture in 1971, and almost all subsequent DDFS were based on this architecture. As shown in Figure 1, the architecture of DDFS mainly consists of two parts: phase accumulator and phase-amplitude converter(PAC). Finally, if an analog signal output is needed, a digital-to-analog converter (DAC) and a low-pass filter (LPF) will be added.

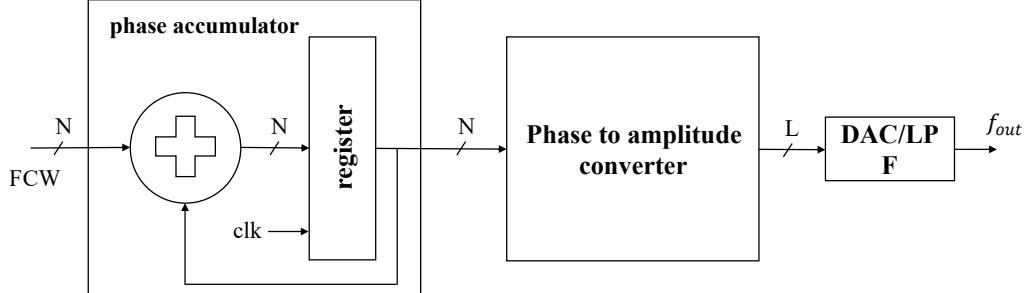

**Figure 1.** General architecture of DDFS.

The frequency control word (FCW) is the input to the entire DDFS and is used to control the frequency of the output signal. The phase accumulator accumulates FCW every

clock cycle to continuously generate new phase stored in the phase register. Its essence is to sample the phase based on the clock frequency, and for each overflow of the phase register, it completes one cycle sampling of the sine signal. The phase is then converted to the corresponding amplitude value in PAC. According to the above description, the frequency of the output signal is determined by the frequency control word (FCW) and clock frequency, which can be expressed as follows:

$$f_{out} = \frac{FCW}{N} * f_{clk} \tag{1}$$

where $f_{clk}$ is the clock frequency, and $N$ is the width of the frequency control word.

The core design of DDFS is the PAC, which can be implemented in many ways, the simplest way is to use a lookup table (LUT), which use the phase as address to index the corresponding amplitude value. The size of LUT increases exponentially with the increase of the bit width N of the FCW, and in order to improve the frequency resolution of the output signal, N is usually taken to a large value, which results in a very large size of LUT, even using the symmetry of the sine function. In [2–5], the phase is truncated to reduce the size of LUT, but it amplifies the error of the PAC and introduces more spurious signals into the output signal. The latest LUT-based DDFS were proposed by Narayan Sinha et al. [6], which use multiplexer-based LUT implemented in a tree structure.

In order to get rid of the limitation of LUT size, many computation-based methods have been proposed. The core idea of these methods is to optimize space complexity with computational complexity. CORDIC algorithm [7] is introduced to calculate the amplitude value in [8–13]. This method calculates trigonometric functions by angle rotation, which can avoid memory access, but the iteration period is longer. Bergeron et al. [14] design DDFS using a multiplier-based angle rotation algorithm and achieve impressive performance. Some work combines the two methods of angular rotation and lookup table to reduce the lookup table size as much as possible while maintaining accuracy [15–18]. In addition, polynomial approximation method is also used for amplitude calculation [19–28]. In polynomial interpolation methods, the first quadrant of the sine function is usually evenly divided into $s$ segments, each of which is approximated separately by a polynomial, and $s$ is usually a power of two. Among these works, Freeman [19] first used 16-segment linear functions to approximate the first quadrant of the sine function, and designed a correction table using ROM. In order to reduce errors, Chebyshev polynomial and Lagrange polynomial interpolation are used to reduce the MAE and mean square error (MSE) respectively, such as [20,21]. The most important performance indicator of DDFS is SFDR. Therefore, many researchers have conducted in-depth studies on SFDR of DDFS based on polynomial interpolation. Langlois et al. [22] theoretically proved the SFDR upper bound of DDFS based on PWL approximation, as shown in Equation (2). Later, Caro [21] proved the SFDR upper bound of DDFS based on first-order and higher-order polynomial approximation, where the SFDR upper bound of first-order linear interpolation is the same as Equation (2). The SFDR upper bounds of second and third order polynomial interpolation are shown in Equation (3) and Equation (4) respectively.

$$SFDR = 16s^2 + 1 \tag{2}$$

$$SFDR = 256s^3 - 96s^2 + 24s - 1 \tag{3}$$

$$SFDR = \frac{5120s^4 + 768s^2 + 5}{3} \tag{4}$$

where $s$ is the number of segments. However, Ashrafi [23] pointed out that the above SFDR upper bound is only true if the polynomial is complete, and analyzed the harmonics generated in the case of even fourth-order polynomial. The polynomial approximation methods mentioned above are all based on uniform segmentation, while there are few works based on non-uniform segmentation, and we only found [24].

No matter what method is used to design PAC, three indicators are the main considerations of the designer: MAE, MSE and SFDR, which indirectly reflect the noise in the signal. The first work proposing the idea of improving sFDR by reducing MAE is [29].

In this paper, we propose a high SFDR DDFS. The characteristic of this DDFS is using a non-uniform PWL approximation method to convert phase into amplitude, rather than the traditional uniform PWL approximation or LUT.We call the method PWLMMAE (Piecewise Linear Minimize Maximum Absolute Error) and will describe it in detail in the next section. This method can calculate the piecewise points, the slope and intercept of each segment, satisfying the preset MAE. Moreover, this method is hardware friendly. Based on this method, We first implemented a set of DDFS under different MAE to study the impact of MAE on SFDR, and then evaluated its hardware overhead. We also investigate the relationship betwween the number of segments in the PWL approximation of sin function and SFDR and compare it with Equation (2). Finally, we implemented multi-core DDFS using a time interleaving scheme to increase the frequency of the output sine signal. In summary, the main contributions of this paper are as follows:

- Proposing a non-uniform PWL approximation method with MAE controlled.
- Implementing a set of DDFS under different MAE based on the above method on FPGA to study the the impact of MAE on SFDR.
- our proposed DDFS breaks through the SFDR theoretical upper bound of DDFS based on the piecewise linear approximation method represented in Equation (2).
- Implementing multi-core DDFS which can achieve 3.9 GHz sampling rate and 114 dB SFDR.

The organization of this paper is as follows. The Section 2 introduces the non-uniform piecewise linear approximation method we proposed, and the Section 3 designs the hardware architecture of single-core and multi-core DDFS based on this method. The Section 4 presents the experimental results, including the approximation of sine function by our PWL approximation method, SFDR of DDFS under different MAE and the performance of multi-core DDFS. Finally, the Section 5 summarizes this paper.

## 2. Non-Uniform PWL Approximation Method

In this section, We first introduce the PWLMMAE algorithm, which is based on [30]. We further reduced the number of segments and mean square error by modifying the way of fitting target curve [31]. The algorithm is used to perform PWL approximation of the first quadrant of sin function, and then a period of sin function is fitted using its symmetry.

### 2.1. PWLMMAE

The core idea of the algorithm is to determine the subinterval straight line by least squares, then calculate the maximum absolute error between the line and the real curve and find the maximum absolute error less than the predetermined error through continuous iteration, the steps of the algorithm are as follows.

1. Input range discretization
   Considering the hardware implementation, the input range should be discretized. For a given input interval $[X, Y]$, the input $x$ should be defined as a vector

$$x = x(1 : NUM) = X, X + \frac{1}{2^Q}, X + \frac{2}{2^Q}, ..., Y \qquad (5)$$

where $Q$ is the number of fractional bits setting in hardware and $NUM$ is the length of the vector.

2. Minimization of MAE for a given width of subinterval

The slope $a$ and intercept $b$ of the subinterval approximation line are first calculated using the least squares method by Equations (6) and (7), and we can use $h(x)$ to represent the approximation line.

$$bn + a \sum x = \sum f(x) \qquad (6)$$

$$b \sum x + a \sum x^2 = \sum x f(x) \qquad (7)$$

where $n$ is the number of discrete points in the subinterval, then, the objective function is denoted by $f(x)$ with $x \in (j : k), 1 \leq j \leq k \leq NUM$, so the error vector can be expressed as Equation (8).

$$\delta = f(x(j : k)) - h(x(j : K)) \qquad (8)$$

The corresponding MAE can also be calculated as

$$MAE = max\{|max(\delta)|, |min(\delta)|\} \qquad (9)$$

To minimize MAE, we shift line $h(x)$ vertically such that $|max(\delta)| = |min(\delta)|$, and the distance $T$ moved can be calculated by Equation (10). The MAE after movement is shown in Equation (11).

$$T = max(\delta) - \frac{max(\delta) - min(\delta)}{2} = \frac{max(\delta) + min(\delta)}{2} \qquad (10)$$

$$MAE = \frac{max(\delta) - min(\delta)}{2} \qquad (11)$$

3. segmentation points

To obtain the maximum segmentation interval, we determine the segmentation points from right to left. Initially, we set START = $x(1)$, END = $x(NUM)$, then perform the PWL method on $x$(START: END) to calculate MAE by Equations (8) and (9). If $MAE \leq E_c$, where $E_c$ is a predefined error, approximation succeeds and set $b = b + T$; otherwise END = END $-$ 1 and repeat above step. Once approximation succeeds, we will update START and END to find the next subinterval, where $MAE \leq E_c$. The values of START and END record the segmentation points, the index $i$ records the number of segments.

Figure 2 shows the process of finding the $i$th segment. $h_1, h_2, ..., h_{i-1}$ represent the approximate lines of the previous $i - 1$ segments respectively, $x(END_{i-1})$ is the start point of the $i$th segment. The end point of first approximation of $i$th segment is $x(NUM)$.Continuously move the end point to the left in steps $\frac{1}{2^Q}$ until the error between the approximate line and the objective curve is less than $E_c$. Then, we can obtain the end point $x(END_i)$ of the $i$th segment and the approximate line $h_i(x) = a_i x + b_i$.

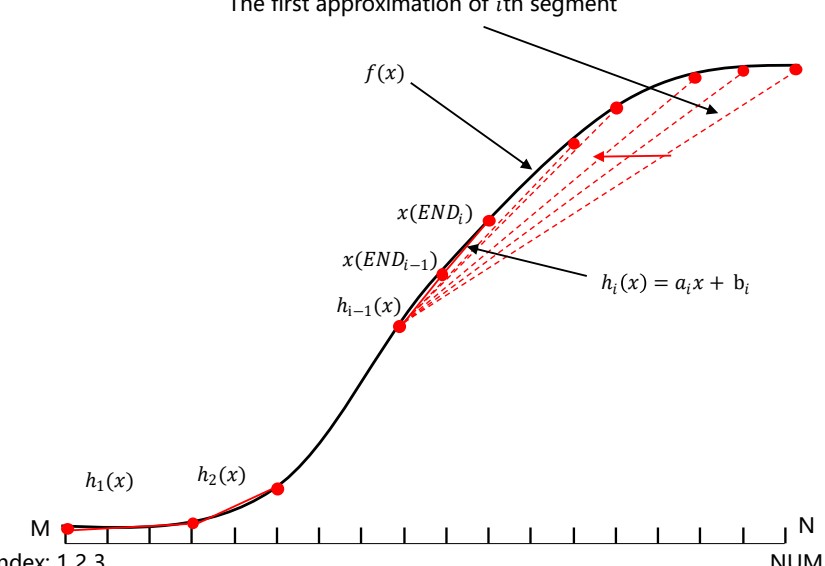

**Figure 2.** The process of seeking for the *i*th segment that meets the error requirements.

In summary, the implementation of PWLMMAE is as follows (Algorithm 1). On the fifth line, *polyfit* is a function that uses the least squares method to fit the curve and the argument "1" indicates first-order.

---

**Algorithm 1:** PWLMMAE

---

1   $x = M : \frac{1}{2^Q} : N, NUM = length(x)$;

2   $j = 1, END = NUM, i = 1, F = 1$;

3   **while** $j! = NUM$ **do**

4      **while** $F == 1$ **do**

5         $a_i, b_i = polyfit(x(j : END), f(x(j : END)), 1)$;

6         $h(x) = a_i * x + b_i$;

7         $\delta = f(x(j : END)) - h(x(j : END))$;

8         $MAE = (max(\delta) - min(\delta))/2$;

9         **if** $MAE \leq E_c$ **then**

10           $b_i = b_i + \frac{max(\delta) + min(\delta)}{2}$;

11           store $a_i, b_i, x(j), x(END)$;

12           $F = 0$;

13         **end**

14      **end**

15      $i = i + 1, j = END, END = NUM, F = 1$;

16 **end**

---

### 2.2. PWL Approximation of the Sin Function

Let's consider the first quadrant of the sin function:

$$f(x) = sin(\frac{\pi}{2}x) \quad 0 < x < 1 \tag{12}$$

Set $E_c = 0.001, Q = 10$. The segmentation results can be obtained through MATLAB R2021a as shown in Table 1.

**Table 1.** Segmentation results of the first quadrant of the sin function.

| i | a | b | End Point |
|---|---|---|---|
| 1 | 1.54503497834031 | 0.000990880661757787 | 0.2001953125 |
| 2 | 1.43612208005773 | 0.0227876190983088 | 0.326171875 |
| 3 | 1.29779670692432 | 0.0679214996797700 | 0.43359375 |
| 4 | 1.14069314244100 | 0.136030419000189 | 0.5302734375 |
| 5 | 0.970322027885993 | 0.226383524844325 | 0.62109375 |
| 6 | 0.790355834163542 | 0.338143580690051 | 0.70703125 |
| 7 | 0.604027272905406 | 0.469880462191978 | 0.7900390625 |
| 8 | 0.412863639934916 | 0.620903909462291 | 0.87109375 |
| 9 | 0.218415511777311 | 0.790288325648528 | 0.951171875 |
| 10 | 0.0602097594473886 | 0.940157596611038 | 1 |

The first quadrant of the sin function is divided into 10 segments, the first segment starts at 0, the end point of the previous segment is the start point of the next segment, and the end of the last segment is 1. Using the symmetry of the sin function, we make a PWL approximation of one period, as shown in Figure 3. The sin function realized by PWLMMAE fits well with the sin function in MATLAB, and MAE is within the preset range, which verifies the effectiveness of the algorithm.

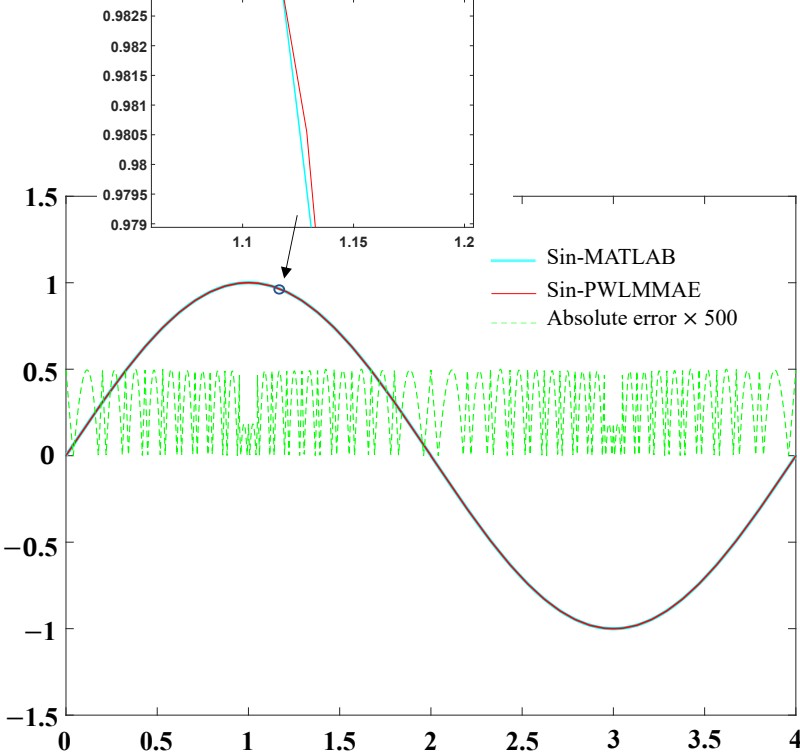

**Figure 3.** PWL approximation of the sin function.

## 3. Hardware Architecture of DDFS

In this section, we focus on the single-core DDFS hardware architecture, and then make some minor changes to it to get a multi-core DDFS architecture.

### 3.1. Single-Core DDFS

Before introducing the overall architecture of DDFS, we discuss the hardware implementation of the PWL approximation. Suppose we obtain the *s* linear segments of the sin function by executing the PWLMMAE algorithm introduced in the previous section in MATLAB, and the start point, end point, slope *a*, and intercept *b* of each segment are

also obtained. We store the *a* and *b* of each segment in two separate LUTs. The hardware implementation architecture of the PWLMMAE algorithm is shown in Figure 4. In order to compute the approximate function value of the input *x*, the first step is to determine in which interval segment the input *x* is located before indexing to the corresponding *a* and *b*. The independent variables in the first quadrant of the sin function are all positive, and both the interval endpoints and the input can be expressed as unsigned numbers, so we use $s-1$ parallel comparators and a multiplexer to implement segment indexing instead of using a subtractor as in [30]. Comparison of unsigned numbers is obviously more efficient than subtraction, and it is sufficient to start comparing from the high significant bits. The input *x* is compared with the first $s-1$ interval endpoints at the same time, if $x \geq x(END_i)$, then $sign_{i+1} = 0$, indicating that the input *x* is not in the ith segment, otherwise $sign_{i+1} = 1$, which does not mean that the input *x* is in the *i*th segment, and the results of the $s-1$ comparators can form an $s-1$ bits vector $\{sign_s : sign_2\}$, which will be used as the input of the multiplexer to select the segment index. The mapping of this vector to segment index is shown in Table 2, where index1, index2 and indexs represent the address or index of the slope *a* and intercept *b* of the first, second and last segment in the LUT.

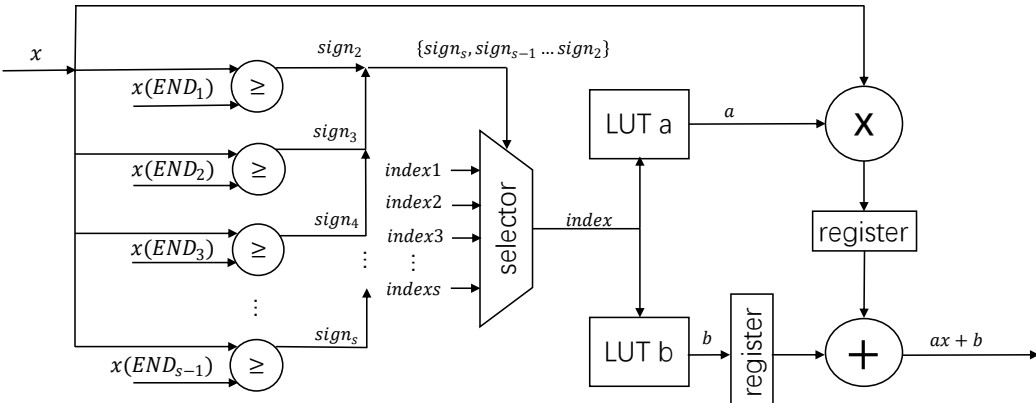

**Figure 4.** The hardware architecture of PWLMMAE algorithm.

**Table 2.** The mapping of the vector to segment index.

| $\{sign_s : sign_2\}$ | Index |
|---|---|
| $s - 1'b111\dots111$ | index1 |
| $s - 1'b111\dots110$ | index2 |
| $\dots$ | $\dots$ |
| $s - 1'b000\dots000$ | indexs |

After getting the segment index corresponding to the input *x*, we can go to the LUT to get the corresponding *a* and *b*, and then perform the multiplication and addition operation, which requires a multiplier and an adder. Since the bit width of input *x* and slope *a* are so long that the multiplication is time consuming, it is necessary to add a register after the multiplier. In this way, the delay for the whole approximate calculation is only one clock cycle.

Now, we design the entire hardware architecture of DDFS based on the hardware architecture of the PWLMMAE algorithm described above. In fact, we only need to replace the PAC in Figure 1 with the PWLMMAE hardware architecture and then add simple control logic to exploit the symmetry of the sin function. The entire DDFS architecture is shown in Figure 5.

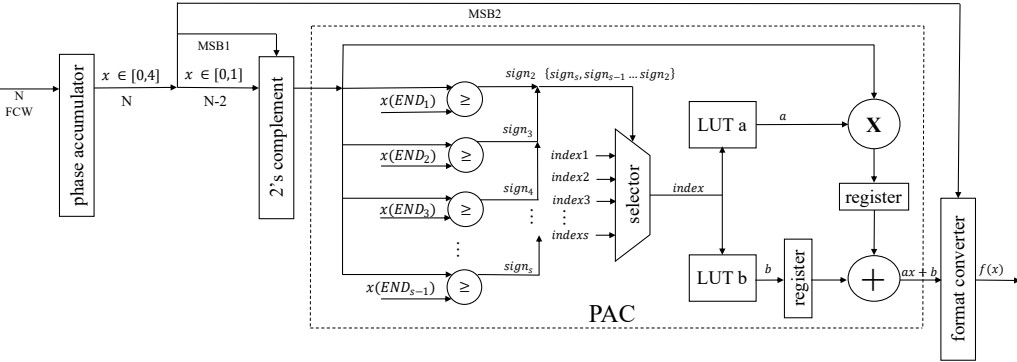

**Figure 5.** The hardware architecture of proposed DDFS.

The input of the entire DDFS is an N-bits FCW, which is accumulated by the phase accumulator to produce an N-bits phase $x$, $x \in [0, 4]$, with the 2 most significant bits being the integer part and the remaining N-2 bits being the fractional bits. In order to take advantage of the symmetry of the sine function, truncation operations and 2's complement operations are required. We truncate the N bits phase to N-2 bits, keeping only the fractional part. The truncation operation maps the phase to the first quadrant by a 2's complement operation. Whether or not to perform the 2's complement operation is determined by the most significant bit (MSB). We illustrate this process with an example below. For example, $x$ = 12'b010111000111, it is in the second quadrant. Its 2 most significant bits are the integer part, the remaining 10 bits are the fractional part, so $x$ is expressed in decimal as 1.4443359375. After the truncation, $x$ = 10'b0111000111 (0.4443359375 in decimal), then take the complement $x$ = 10'b1000111001 (0.5556640625 in decimal). By the symmetry of the sin function, $sin(\frac{\pi}{2} * 1.4443359375) = sin(\frac{\pi}{2} * 0.5556640625)$. Thus we can get the approximate value of $sin(\frac{\pi}{2}x)$ in the interval [0,2]. Similarly, we can obtain the absolute value of the approximation of $sin(\frac{\pi}{2}x)$ in the interval [3,4]. The approximate values obtained also need to be converted to the correct encoding format in a format converter, not the simple one with signed numbers.

### 3.2. Multi-Core DDFS

Multi-core DDFS [6] is able to increase the sampling rate and the frequency of the output signal. Its main idea is to use different cores to calculate simultaneously the values of functions corresponding to different offset phases. Its architecture is shown in Figure 6.

There are two main differences between the multi-core architecture and the single-core architecture. The first one is that the input to each core is the MFCW (multicore frequency control word), which can be calculated from Equation (13). The another difference is the addition of an adder after the phase accumulator for adding the phase offset which can be represented by Equation (14) for each core between the different cores.

$$MFCW = M * FCW \tag{13}$$

$$offset_j = j * FCW \tag{14}$$

where $M$ is the number of cores, $j$ denotes the $j$th core, and $1 \leq j \leq M$. With such an architecture, the sin function can be sampled M times in one clock cycle, increasing the sampling rate by a factor of $M$.

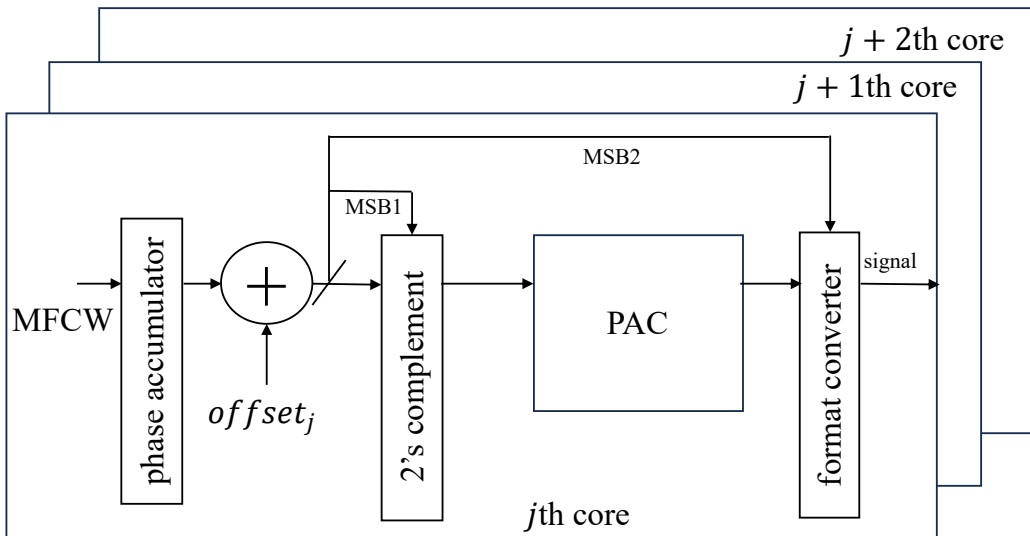

**Figure 6.** The Hardware architecture of multi-core DDFS.

## 4. Experimental Result

This section describes the experimental setup of our proposed DDFS and the experimental results. The DDFS is implemented using the Verilog hardware description language and deployed to the Xilinx AXU15EG development board after simulation and synthesis by the Vivado tool. The input bit width N is uniformly set to 12, and the output data is read using ila IP cores, followed by MATLAB for spectrum analysis. The performance of the single-core DDFS is shown first, and then the experimental results of multi-core DDFS is introduced.

### 4.1. Performance Evaluation of Single-Core DDFS

We have implemented six sets of DDFS under different MAEs, and the spectrum images of the output signals are shown in Figure 7. The FCW and clock frequency of DDFS in these images are the same, FCW = 200, clock frequency is 200 MHz. But the bits of output signal are different under different precision. The bit number of output signal in Figure 7a–c is 16 bits, and that in Figure 7d–f are 17 bits, 19 bits, 20 bits respectively. From the spectrum images, we can discover that the SFDR of DDFS increases and the noise floor reduces with the increase of the approximate accuracy, which means our idea of improving the SFDR by minimizing the MAE is feasible.

Next we discuss the relationship between the number of linear segments s and the SFDR. We implemented six sets of DDFS with different MAE corresponding to different numbers of linear segments, and the specific number of segments and SFDR are shown in Table 3. Each set of DDFS generate three signals at low frequency (FCW = 200), medium frequency (FCW = 1000) and high frequency (FCW = 1800), and their SFDR is measured separately and averaged. As introduced in Section 1, in [21,22], Caro and Langlois derived theoretically the upper bound of SFDR for DDFS using uniform segmented linear approximation, respectively, whose upper bound of SFDR is shown in Equation (2) and can be expressed in Equation (15) after converting to decibels.

$$SFDR = 20\log(16s^2 + 1) \approx 20\log(16s^2) = 24 + 40\log s \qquad (15)$$

This equation is derived through Fourier transform. During the derivation process, an important prerequisite is uniform segmentation, which means that the segmentation points can be represented using the number of segments $s$. In our proposed non-uniform PWL method, this prerequisite is not valid and mathematical derivation is not feasible. Therefore, we directly tested SFDR under different segmentation numbers through experiments.

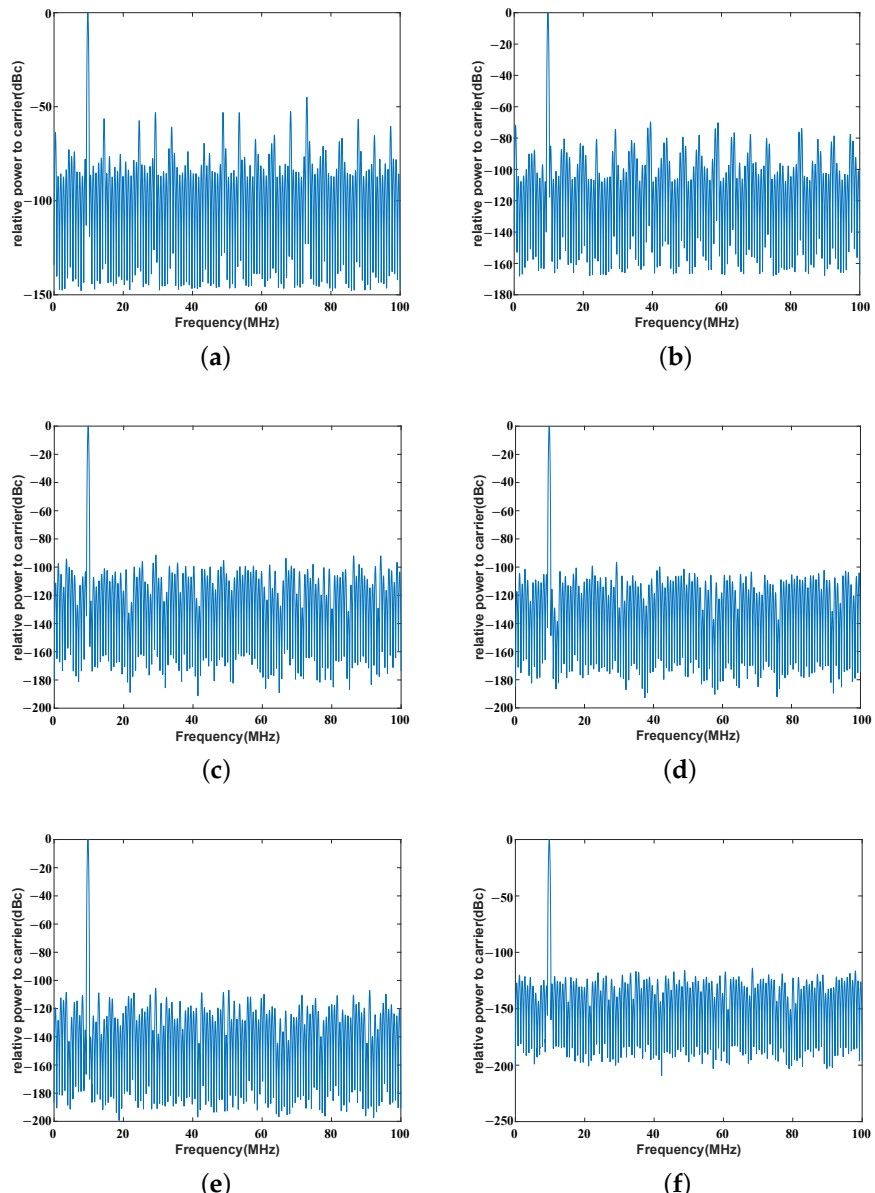

**Figure 7.** Spectrum of DDFS output signal under different MAE, FCW = 200, clock frequency is 200 MHz. (**a**) MAE = 0.01 (**b**) MAE = 0.001 (**c**) MAE = 0.0001 (**d**) MAE = 0.00005 (**e**) MAE = 0.00002 (**f**) MAE = 0.00001.

Setting $s$ to the power of 2, the equation is plotted as an image as shown in the blue curve in Figure 8. And the red curve in the figure shows the relationship between the number of segments and the SFDR of the DDFS using PWLMMAE segmentation method. This experimental result shows that the DDFS based on our method has a higher upper bound of SFDR than the DDFS based on the uniform segmentation linear approximation method in [21,22]. Explaining from the perspective of MAE, for a fixed interval, the more the number of segments, the higher the accuracy of the approximation and the smaller the MAE. Our proposed PWLMMAE method has a higher approximation precision with the same segments.

**Table 3.** Averaged SFDR of DDFS under different MAE.

| MAE | Segments | SFDR (dB) |
|---|---|---|
| 0.01 | 4 | 44.94 |
| 0.001 | 10 | 69.64 |
| 0.0001 | 31 | 91.41 |
| 0.00005 | 44 | 96.58 |
| 0.00002 | 70 | 105.47 |
| 0.00001 | 100 | 114.04 |

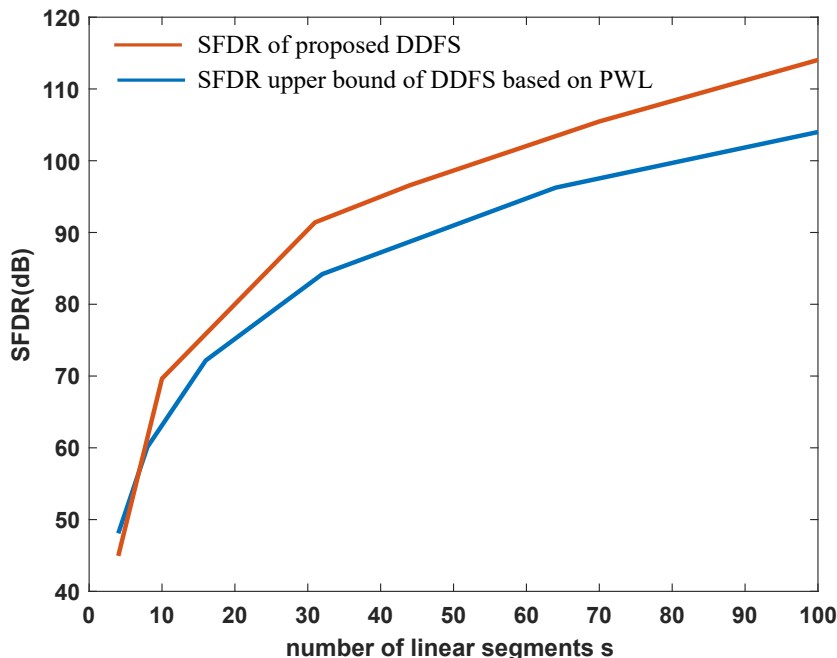

**Figure 8.** The relationship between SFDR and the number of segments *s* of DDFS based on different PWL approximation method. The red curve represents the SFDR measured by the proposed DDFS, while the blue curve is the upper bound of the SFDR inferred in [21,22].

In the following, we comprehensively evaluate the performance and resource consumption of DDFS. For performance we mainly consider SFDR and maximum clock frequency, because SFDR reflects the quality of the output signal, and maximum clock frequency reflects the highest frequency of the output signal. The resources used include LUT, Flip Flop (FF), and DSP. As can be seen in Figure 9, as the MAE decreases, the SFDR of DDFS will increase, but the hardware resources required will also be more, and the maximum clock frequency also tends to decrease, which is mainly caused by the increase in the number of segments.

Table 4 shows the performance of this work compared to other FPGA-based DDFSs. the DDFS with MAE = 0.00001 uses 399 LUTs, 66 FFs and 3 DSPs and is able to achieve a SFDR of 114.04 dB with a maximum clock frequency of 244 MHz. SFDR is the most important indicator, reflecting the quality of the DDFS output signal. Except for [14], our DDFS is leading in SFDR without significant increase or even decrease in resources. Compared to the latest work [6] to the best of our knowledge, our work achieves a 4 dB higher SFDR. We use 67 more LUTs and three more DSPs, while we use 150 fewer FFs. The SFDR of this work is 41.84 dB higher than [9], but uses significantly 99 less LUTs and 140 FFs. Compared to [12,25], our proposed DDFS has significant advantages in SFDR, clock frequency, and resource consumption. It is foreseeable that the SFDR will be further improved if we further reduce the MAE.

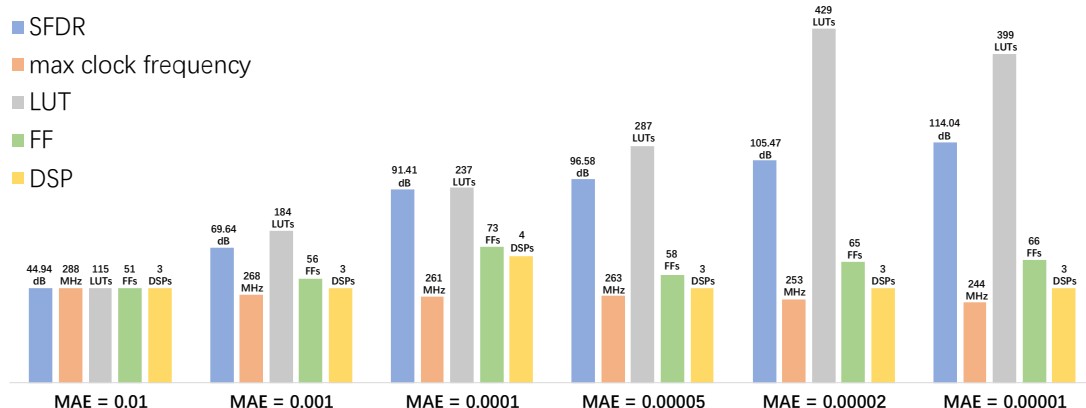

**Figure 9.** Comprehensive evaluation of DDFS under different MAE.

**Table 4.** Performance comparison among different DDFSs.

| DDFS Design | SFDR (dB) | Max Clock Frequency (MHz) | Target Device | Output Bits | Resource Utilization |
|---|---|---|---|---|---|
| Proposed | 114.04 | 244 | AXU15EG | 20 | 399LUTs, 66FFs, 3DSPs |
| [6] | 110 | 250 | Artix-7 | 16 | 332LUTs, 216FFs |
| [9] | 72.2 | 251 | Virtex-6 | 16 | 498LUTs, 206FFs |
| [12] | 96.31 | 107.216 | XC3S500E | 16 | 967LUTs, 788FFs, 487 slices * |
| [14] | 120 | 1000 | Virtex-7 | 16 | 46slices, 3DSPs |
| [25] | 95 | 192 | Spartan-2 | 16 | 566slices |
| [32] | 104.1 | 281.7 | Virtex-5 | 16 | 158slices |

\* Taking xilinx7 series FPGAs as a reference, a slice contains four 6-input LUTs, eight FFs, and some multiplexers and carry logic.

### 4.2. Performance of Multi-Core DDFS

According to the analysis of the performance and resource consumption of proposed DDFS in the previous subsection, we choose the DDFS with MAE = 0.00001 to implement the 16-core DDFS, which increases the output signal frequency while maintaining high SFDR. The sampling rate of single-core DDFS is 244 MHz, so the sampling rate of 16-core DDFS can achieve 3.9 GHz. Figure 10 shows the spectrum of the output signal of multi-core DDFS with FCW = 1800, the output signal frequency is about 1.7 GHz, and the SFDR can reach 114.04 dB.

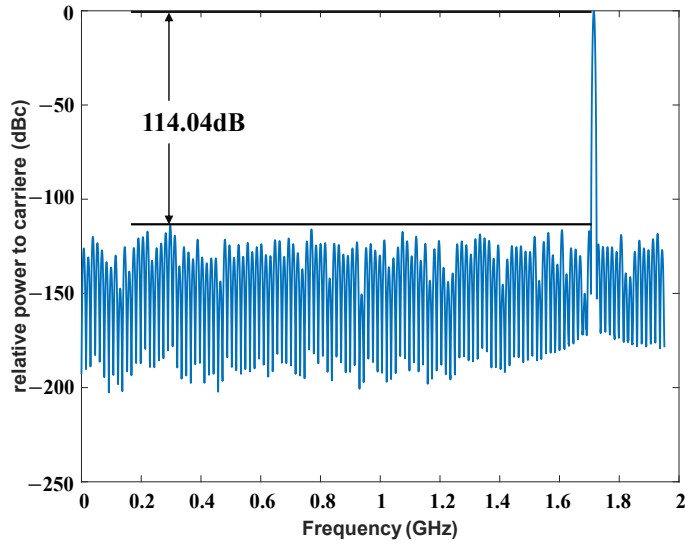

**Figure 10.** Spectrum of 16-core DDFS output signal.

## 5. Conclusions

In this paper, we propose a DDFS based on a non-uniform segmented linear approximation method. We implement a set of single-core DDFS with different MAE, and benefitting from the segmentation algorithm we use, our proposed DDFS has a higher SFDR upper bound compared to other segmented linear approximation based DDFS, breaking the theoretical upper bound obtained by previous researchers. Compared with other FPGA-based works, our work can achieve higher SFDR without significantly increasing or even decreasing the resource consumption. finally, considering the DDFS performance and resource consumption, we choose a DDFS with MAE = 0.00001 to implement a multi-core DDFS, enabling a sampling rate of 3.9 GHz and a SFDR of 114.04 dB.

**Author Contributions:** Methodology, X.L.; validation, T.Z. and Y.P.; writing—original draft preparation, X.L.; writing—review and editing, X.L. and T.Z.; supervision, L.Z. and X.H. All authors have read and agreed to the published version of the manuscript.

**Funding:** The Opening Foundation of State Key Laboratory of High-Performance Computing, National University of Defense Technology, under Grant No. 202201-05.

**Institutional Review Board Statement:** Not applicable.

**Informed Consent Statement:** Not applicable.

**Data Availability Statement:** The data presented in this study are available on request from the corresponding author. The data are not publicly available due to privacy

**Conflicts of Interest:** The authors declare no conflict of interest.

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
