# Peer review of "FPGA Implementation of a Higher SFDR Upper DDFS Based on Non-Uniform Piecewise Linear Approximation"

_applsci, doi:10.3390/app131910819_

Round 1

Reviewer 1 Report

This article proposes a piecewise Linear (PWL) approximation method to design a direct digital frequency synthesizer (DDFS), and implement the DDFS on FPGA. This method uses non-uniform PWL approximation to approximate the first quadrant of sin function, and the expected maximum absolute error (MAE) can be achieved with the minimum number of segments. The SFDR can reach 114dB using 399 LUTs, 66 flip flops 10 and 3 DSPs. In general, the article is well organized, some concerns need to be addressed, which are listed below:

(1) The novelties and significance of the DDFS design are suggested to be further highlighted.

(2) In the comparison table, the quantitative advantages of the design are suggested to be illustrated more clearly.

(3) The organization can be further optimized.

Reviewer 2 Report

It is necessary to express the innovation of the presented work more precisely and clearly to distinguish it from previous works.

The flowchart presented in Figure 2 can be redrawn more clearly and by expressing some sentences.

How much has the accuracy of the presented work increased compared to similar works? This point should be carefully examined from other perspectives as well.

How fault-tolerant is the presented method? Can you provide an analysis in this regard?

What is the sensitivity of this method to the frequency of the clock signal and the sinusoidal signal?

 Author Response

Reviewer 3 Report

Dear Author

I have reviewed your paper, "FPGA Implementation of a Higher SFDR Upper DDFS Based on Non-uniform Piecewise Linear Approximation, " which is well-written and formatted. Even though some of the points are against the manuscript's quality, that are listed

1.      In the abstract, the technique is missing

2.      if Figure 1 is cited from other references. Please cite the reference ... Example: Figure 1. The general architecture of DDFs [1]

3.      The flow chart (Figure 2) explanation is not technical

4.      Figure 4: How do you select the index, and how many inputs are given? and what will be the selection input

5.      Figure 7 is not explained properly, the author must explain how it is different from existing circuits in terms of all efficiency

6.      The comparison table (4) must express the superiority of your proposed circuits

7.      I think Figure 10 must be removed from the conclusion… The conclusion should be as text word only.

8.      The comparisons reference [6 and 9] only recent journals. All other references are a bit old. Please compare within 5 years.

Round 2

Reviewer 2 Report

The Manuscript can be accepted.

Reviewer 3 Report

Dear Author

Greetings

Kindly inform you that, the author has addressed all comments